# Water–Energy–Nutrients Nexus of Urban Environments

**Armando Silva-Afonso** [1,2,3] and **Carla Pimentel-Rodrigues** [1,3,4,*]

1   RISCO—Research Center for Risks and Sustainability in Construction, Campus Universitário de Santiago, University of Aveiro, 3810-193 Aveiro, Portugal; asilva.afonso@outlook.pt
2   WFEO—World Federation Engineering Organization (Committee on Water), Maison de l'Unesco, 1, rue Miollis, 75015 Paris, France
3   ANQIP—National Association for Quality in Buildings Services, Campus Universitário de Santiago, University of Aveiro, 3810-193 Aveiro, Portugal
4   ISCIA—Higher Institute of Information and Administration Sciences, 3810-488 Aveiro, Portugal
*   Correspondence: anqip@anqip.pt

**Abstract:** The objective of this article is to deepen knowledge about the existing connections, at the level of urban environments, between energy, water, and nutrients (or food). Energy and basic resources—water and food—are closely interconnected, which is why the water–energy–food nexus constitutes the essential integrated approach to ensuring the sustainable development of humanity. This nexus is also valid in urban environments and can be adapted for buildings, interconnecting, in this case, water, energy, and nutrients. This article is a literature review in this area, intending to highlight the strong connections between water, energy, and nutrients at the level of buildings, integrating the results obtained in different studies and showing the global importance of this nexus. The water–energy relationship in buildings is already well known in terms of the production of domestic hot water or building pumping, for example, but it turns out that it goes far beyond this interrelationship, also having implications for public networks. Regarding the water–nutrients nexus in urban environments, it can play an important role in terms of food security for humanity, especially regarding the possibility of recovering phosphorus in buildings.

**Keywords:** water–energy nexus; water–energy–nutrients nexus; net-zero buildings





## 1. Introduction

The increasing need for the sustainable use of energy and resources globally, as a result of population growth on our planet, economic development models, and, in some regions, climate change, requires a holistic and integrated approach as a guiding principle of development policies [1–5]. This principle is highlighted in the UN Agenda 2030, where it is stated that it is through this integrated approach that cross-connections can be promoted among social progress, economic development, and environmental protection [6–9].

Energy and basic resources—food and water—are closely interconnected, which is why the water–energy–food nexus constitutes the essential integrated approach to ensure the sustainable development of humanity. In general terms, water is crucial in most forms of energy production and energy is necessary for many uses of water, such as supplying populations or heating domestic hot water [10]. Energy and water are also closely linked to food. In the current conditions of the planet, with a growing global population, agriculture is responsible for 70% of total freshwater consumption [11], while food production and its distribution chain represent around 30% of all energy consumption [12].

This water–energy–food nexus can also be valid in urban environments [13–16] and can be adapted for buildings, particularly in their use phase [17]. In this case, this nexus must be seen concerning the interconnection between water, energy, and nutrients (or fertilizers).

It may be added that some authors, noting that urban agriculture is expanding rapidly to meet the growing demand for locally produced food, propose an intermediate concept,

dividing the "food" component in urban environments into nutrients and food, that is, considering a water–energy–nutrients–food nexus [18]. This extended nexus, however, is only of interest in the analysis of resource stocks and flows when there is urban agriculture, excluding other situations, such as the use of nutrients recovered in the fertilization of green roofs or urban gardens.

Due to sustainability concerns, net-zero-energy buildings (NZEBs) have already become a reality in many areas of the planet, and the next actions should be the design and dissemination of net-zero-water and net-zero-nutrients buildings. However, this approach should not be conducted independently. In fact, "net-zero-buildings" should be the constructive solutions of the future, but they should consider the intrinsic relationship that also exists in urban environments between water, energy, and nutrients.

The net-zero-building concept intertwines with the circular use of resources, but these concepts are not equivalent. In the case of energy, net-zero-energy buildings do not mean the circular use of the resource, but rather that the total amount of energy used by the building is approximately equal to the amount of renewable resources produced or available locally.

In relation to water, part of this resource can be used circularly, through water recycling, but alternative renewable sources, such as rainwater, can also be considered. In the case of buildings with zero nutrients, the use of resources can be circular [19], as some nutrients can be recovered and used as fertilizers in green roofs or urban agriculture, as is the case with phosphorus (P), as illustrated in [20–22].

When there is a centralized drainage system, the recovery of nutrients discarded in buildings can be performed downstream in wastewater treatment plants (WWTPs), for example [23–27]. For this solution, several technologies have been proposed over the last few years [28–35], which will be referred to later, although none of them have become widespread or have been recognized, to date, as being of obvious interest from a technical–economic perspective.

This article aims to highlight, at the level of urban environments, the strong connections between water, energy, and nutrients. Based on a review of some recent publications in this area, we intend to integrate the results obtained in partial studies related to each of these interconnections and to show the importance of the nexus between energy, water, and nutrients in buildings.

## 2. Materials and Methods

For the literature review, keywords related to the nexuses of water–energy, energy–nutrients, and water–nutrients in buildings were used, as well as keywords related to the global water–energy–food and water–energy–nutrients nexuses. Articles on "net-zero buildings" were also analyzed. It was found that the available literature on the relationship between water, energy, and nutrients in buildings is relatively restricted and focuses mainly on isolated analyses of each of these resources or the relationship between water and energy, specifically the production of sanitary hot water (SHW). The concepts of net-zero-water or net-zero-nutrients buildings have also been the subject of few studies. A search of the scientific literature centered on these concepts returned a relatively small number of articles, unlike net-zero-energy buildings (NZEBs), also called "nearly zero-energy buildings", where extensive references can be found.

However, it appears that most studies on NZEBs focus exclusively on solutions to reduce the energy needs of the network through better thermal insulation, self-production, increased efficiency in use, etc., without addressing the possible implications outside the building or interconnections with other resources, such as water, except for SHW production.

In this article, the links between water nutrients, energy nutrients, and water–energy will be analyzed separately. The latter will focus not only on the traditional approach to SHW production but also on reducing energy consumption in public water supply and drainage systems associated with greater efficiency in water cycle management for buildings.

## 3. Results

### 3.1. Water–Nutrients Nexus in Buildings

Regarding the water–nutrients nexus in buildings, toilets and urinals, when they exist, are essential components in the relationship between these resources. These devices have always played a key role in the context of the water–nutrients nexus and have high potential, in the future, to make important contributions to the promotion of solutions for the sustainable use of nutrients at the scale of buildings and their surroundings. In fact, nutrients recovered in buildings, from manure, urine, or domestic effluents, can be used in urban agriculture or the fertilization of public gardens or green roofs.

In early civilizations with sanitation concerns, such as Rome, some buildings were equipped with direct flush solutions, such as open seats over water lines. Pit latrines appeared later, usually in rural areas (Figure 1), and were improved and reinvented over time, allowing the recovery of nutrients and their subsequent use in agriculture with progressive health safety [36,37]. It can be said that these latrines thus constituted the first sustainability solution within the water–nutrients nexus in buildings and their surroundings. (Even in cases where they were not equipped with flushing devices for cleaning, water was always present, as it represents 95% of urine and 75% of feces).

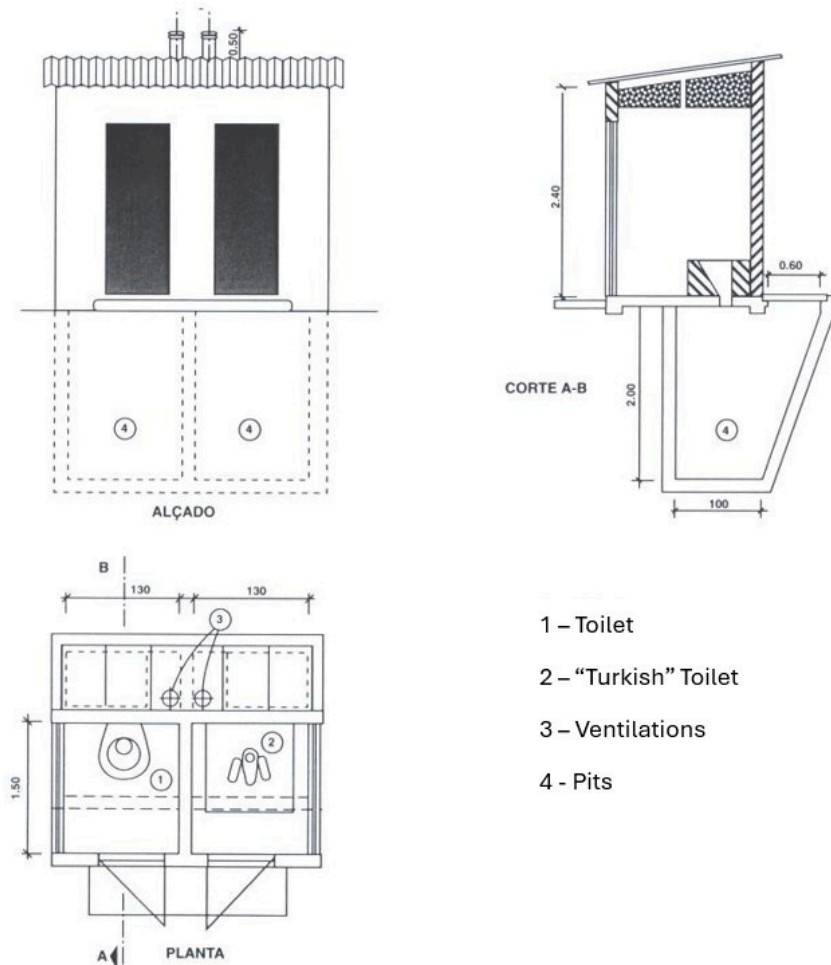

**Figure 1.** Pit latrine [38].

Recently, in parallel with concerns about water efficiency in toilets, the importance of resource recovery has resurfaced, leading to the development of solutions such as composting toilets. Examples of solutions of this type include the toilets installed in the Choi building at the University of British Columbia, which do not require water or the Aquatron system [37]. The latter system can operate with less than 3 L per flush

and allows the separation of solid and liquid waste, allowing the separate treatment of these two types of waste, with composting in the case of solid waste. More recently, the nanomembrane toilet, which is also a waterless solution, was developed by researchers at Cranfield University and promoted by the Bill and Melinda Gates Foundation [37].

Food production requires a total of seventeen elements, among which nitrogen, phosphorus, and potassium are considered the fundamental ones (macronutrients). These three elements can be found in human urine, which also contains some secondary micronutrients.

However, special attention must be paid to phosphorus and the possibility of its recovery. There are several warnings about the urgency and importance of this recovery, from, for example, the European Commission (Parliament statement of 24 May 2012 (§52)). Its scarcity could compromise humanity's food security, but population growth, wars, and the increase in global agriculture are intensifying the pressure on the finite reserves of this nutrient. About 90% of phosphorus reserves are in China, Russia, the United States, and Morocco, but it is estimated that the reserves whose extraction will be economically viable will be exhausted in the next 30–40 to 300–400 years. The wide range of this estimate arises from doubts regarding the volume and quality of reserves and uncertainties regarding future demand [20–22].

On the other hand, the discharge of domestic and industrial effluents rich in phosphorus into the water environment is the main cause of eutrophication, which is a very significant problem affecting the quality of freshwater bodies that has yet to be adequately resolved, at least in Europe. As the recovery of phosphorus is an emergency in the face of food safety and pollution problems, it appears that its elimination through urine is one of the main contributors to its dissemination in the environment and loss in the value chain. Indeed, urine contributes almost half of the total P load in wastewater.

An average adult excretes a significant amount of phosphorus through urine (around 1 g per day), but there are still no established solutions for its recovery in water bodies. Phosphorus recovery in urban wastewater treatment plants (WWTPs) is theoretically possible and has been the subject of several studies and pilot installations, proposing different technologies, such as infiltration–percolation processes [39], microalgae [40,41], membrane contactors [42], ion exchange processes [43], selective electrodialysis [44], simultaneous nitrification–denitrification [45], electrochemical pH modulation [46], and others [47–54]. However, recovery at source, that is, in buildings, presents clear advantages in reducing dilution and minimizing costs and energy consumption when compared to recovery processes in treatment plants.

Usually, urine recovered from buildings can be directly used as fertilizer without the need for a specific process of separation and recovery of phosphorus (or other nutrients). The biggest problem with using urine directly as a fertilizer, which should be avoided, is the possible presence of enteric pathogens. In general, human urine does not contain pathogens, but fecal cross-contamination can occur during the separation and collection process. Storing source-separated urine in sealed containers is considered the simplest approach to reducing the number of pathogens. Several studies have demonstrated that a storage time of a few months is sufficient to reduce the risk of transmission to below the acceptable limit [55]. During this maturation phase, urea is hydrolyzed into ammonia, which causes the pH to increase from approximately pH = 6 to pH = 9 after a few days or weeks. At this point, ammonia represents 90% of the nitrogen in stored urine, and it is this harsh chemical environment that allows the sanitation of source-separated urine [55]. In some specific situations, other problems may exist, particularly when the presence of medicines and other micropollutants in urine intended for fertilization applications is expected [56,57].

It can be noted that, on green roofs for example, fertilizations are typically performed twice a year. This periodicity is perfectly compatible with the periods necessary for urine maturation, which must be at least 2 to 3 months. Even if other fertilization periods are necessary given the type and age of the species, it does not appear that incompatibilities will arise in these applications.

The direct use of matured urine for agricultural purposes has already been the subject of pilot projects in several countries, such as South Africa, Germany, the Netherlands, Sweden, and China, which has already installed more than 700,000 urine-diverting toilets since the end of the last century [37,58].

In Sweden, urine diversion in toilet facilities has been considered a good solution for rural areas to reduce nutrient enrichment in natural waterways, with more than 135,000 urine diversion toilets installed. There are also specific recommendations for the use of urine collected in buildings [37]. In the Netherlands, the authority responsible for water management in the city of Amsterdam developed a pilot program for collecting urine in public toilets, estimating an annual production of 1000 tons of fertilizer, to subsequently fertilize public gardens and green roofs.

Although current efforts focus mainly on the recovery of urine and its direct use as fertilizer after maturation, technologies are also being developed for the direct recovery of phosphorus from urine at the source, based, for example, on physical separation processes with nanotechnological structures of alumina [59,60]. In any case, urine separation in buildings, whether or not one is aiming at the subsequent recovery of phosphorus, requires a revolution in current bathrooms, with the generalization of the installation of urine-diverting toilets and the generalization of the installation of urinals in residential buildings, including specific models for women. These measures may involve new designs for sanitary appliances and installations and, eventually, new sizing rules for building drainage networks.

The development of urine-diverting toilets has been ongoing for several years [61]. However, the acceptance of this type of toilet has been conditioned by problems not yet fully resolved related to the complex maintenance required to avoid the blockage of pipes caused by the precipitation of salts and minerals present in urine, which causes the formation of crystals [62,63].

One of the most recent projects in this context is being developed in Portugal and is in the final testing phase. It is a separating toilet with a traditional exterior design (Figure 2) but with an innovative solution for urine separation, which involves diluting the urine in a small volume of flush water. This condition required specific research on urine maturation to evaluate the efficiency of the separation process and health safety, in turn confirming the feasibility of the later use of matured urine as fertilizer. Through microbiological analyses already carried out, it was confirmed that, after a maturation period of 2 to 3 months, sanitary conditions compatible with the intended applications were achieved [55].

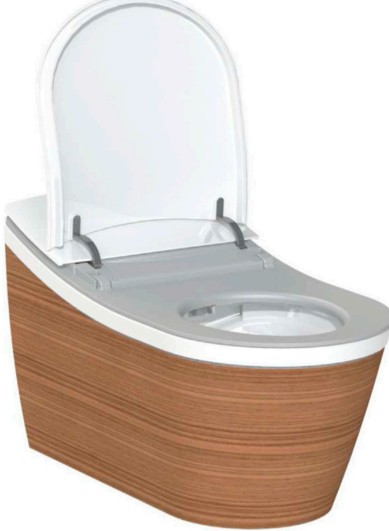

**Figure 2.** Urine-diverting toilet (OLI Company) [57].

From the perspective of the water–energy–food nexus, nitrogen and potassium are equally relevant although without the urgency for recovery that is attributed to phosphorus [64–68]. The contribution of buildings, in these cases, may not be as significant.

### 3.2. Energy–Nutrients Nexus in Buildings

Electric toilets, which use electricity to process waste for nutrient recovery, can be representative of the nutrients–energy nexus in buildings. This is the case, for example, of the composting toilets installed in the C.K. Choi Building, a demonstration building with innovative sustainable design features located at the University of British Columbia (Vancouver) [69].

Energy may also be necessary for the recovery of urine, its maturation, and its subsequent use as fertilizer, for example in pumping equipment. However, the nexus between energy and nutrients will be less relevant than the nexus between energy and water at the building level, in relative terms, given the smaller amounts of energy involved in the former.

In the context of the energy–nutrients nexus, it is interesting to note that the energy consumption for the industrial production of one kilogram of phosphate fertilizer is around 8 MJ, according to industry data. In the case of nitrogen fertilizers, the value is 40 MJ [70]. Furthermore, the fertilizer distribution chain (including processing, packaging, and transport) also consumes significant amounts of energy.

These values show that, in a global analysis, the simple recovery and use of nutrients in buildings and their surroundings can lead to significant reductions in energy consumption. In fact, the local use of urine as fertilizer in buildings or their surroundings only requires small pumps to transfer liquids between containers, and long-distance transportation is not necessary. In other words, energy consumption is negligible.

### 3.3. Water–Energy Nexus in Buildings

In the context of improving water efficiency in buildings, studies show that simply replacing traditional equipment with more efficient devices can lead to significant water savings in buildings, with rapid economic return [71]. This should, therefore, be the solution that should be implemented as a priority, due to its relatively low cost, the significant savings that can be obtained, and the effects it can also have on reducing energy consumption.

A relevant contribution to reducing energy consumption in buildings can result from the application of water efficiency measures, although this aspect is not generally perceived in the implementation of policies to improve energy efficiency in the built environment. Water efficiency has significant consequences in reducing energy consumption in buildings, by reducing energy needs for heating sanitary hot water (SHW) and for possible water pressurization, also reflecting on energy consumption in public networks and of water supply and drainage, by reducing the volumes abstracted, pumped, and treated [71].

A study by the Portuguese Association for Quality in Building Installations (ANQIP) carried out in Aveiro, a medium-sized Portuguese city (70,000 inhabitants), concluded that water consumption in the city could potentially be reduced by around $3.1 \times 10^6$ m$^3$ per year, compared to the current scenario, through the widespread application of more efficient devices in buildings (showers, taps, etc., with category "A" of the ANQIP labeling system for the water efficiency of products) (Figure 3). According to the same study, this reduction, which represents around 24% of the total water consumption in the domestic sector in the city, translates to a reduction in energy consumption of approximately $11.6 \times 10^6$ kWh per year, only concerning the production of SWH in buildings for taps and showers [71].

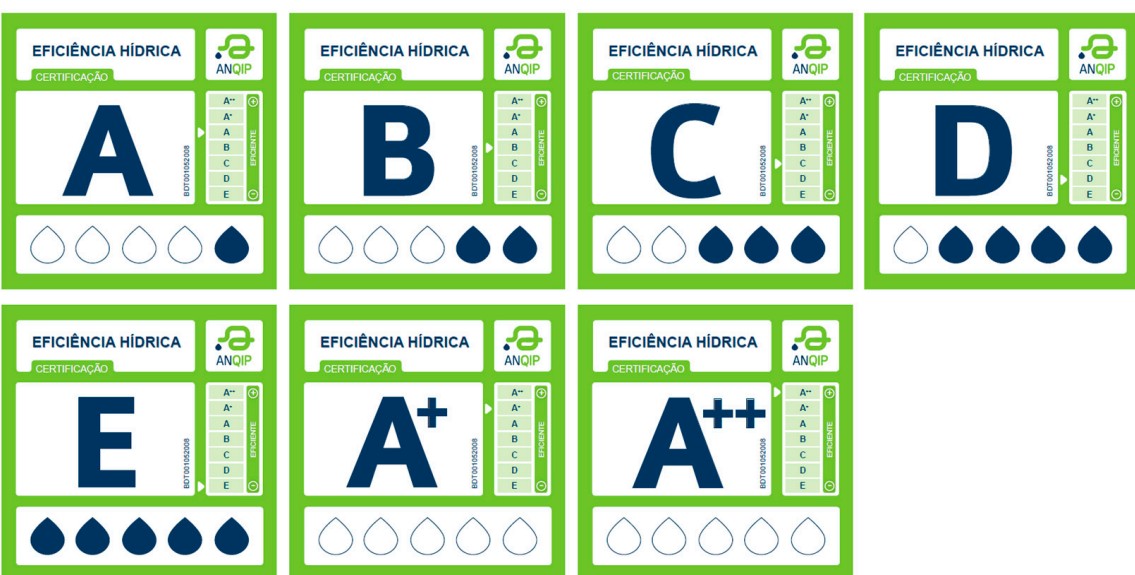

**Figure 3.** ANQIP labelling scheme for water efficiency of products.

According to the Aveiro Water Authority (AdRA), energy consumption is 1.156 kWh/m$^3$ in water production and distribution and 0.818 kWh/m$^3$ in wastewater drainage and treatment. Therefore, in addition to direct energy savings in SHW heating, reducing water consumption in buildings also implies reductions in energy consumption in public water supply and drainage systems (including treatment plants), which can be estimated at $4.4 \times 10^6$ kWh per year [71].

In total values, the potential for reducing energy consumption in the Municipality of Aveiro, as a result of applying simple water efficiency measures in buildings, is around $16 \times 10^6$ kWh per year. This amount reflects the relevance of the water–energy nexus in buildings.

Reducing energy consumption can also translate to a reduction in greenhouse gas emissions, mainly $CO_2$. In Aveiro, the form of energy used in public networks is electricity, which is also the form of energy used in the vast majority of buildings. Considering the indicator available at the time of the ANQIP study, which results from the energy mix used in electricity production in Portugal ($\approx$270 g $CO_2$/kWh), it can be estimated that a reduction in water consumption in buildings of $2.24 \times 10^6$ m$^3$ per year will allow a reduction in emissions of around 4500 tons per year [71].

This result clearly shows the important link between water and energy that exists in buildings. However, this nexus can also be found in other installations or equipment in buildings, and not just in SHW heating or in the associated consumption in public networks. This is the case, for example, of water pressurization systems or vacuum drainage systems, when they exist.

Regarding water-using products (WuPs) in buildings, in addition to concerns about reducing flows or volumes, there has been a rapid evolution in recent decades in their designs, seeking to increase comfort, hygiene or sustainability parameters. This evolution has implied, in many situations, an increase in energy consumption in sanitary installations.

As an example, Table 1 presents a summary of the main recent innovations in toilets, according to manufacturers' catalogues (including smart toilets, which combine the function of a normal toilet with the cleaning functions of a bidet), indicating those that involve energy consumption. It can be observed that most innovations involve energy consumption.

**Table 1.** Recent innovations in toilets and energy needs.

| Recent Innovations in Toilets with Energy Needs | Recent Innovations in Toilets without Energy Needs |
| --- | --- |
| - Touchless flush button;<br>- Seat with motion-activated automatic opening/closing;<br>- Heated seat;<br>- Adjustable seat temperature;<br>- Air purifier deodorization;<br>- UV sanitizing nozzle;<br>- Oscillating jet;<br>- Adjustable water pressure;<br>- Adjustable water temperature;<br>- Air dryer;<br>- Adjustable air dryer temperature;<br>- "Hands-free" remote control;<br>- "Touch screen" remote control;<br>- Ambient lighting;<br>- Multiple memorized profiles;<br>- Speaker integration (radio, music, etc.);<br>- Integrated sensors that alert you to possible leaks;<br>- Seat height adjustment. | - Dual flush;<br>- Discontinuous flush;<br>- Tornado flush;<br>- Rimless pan design;<br>- Soft-close seat;<br>- Self-cleaning nozzle;<br>- Removable nozzle;<br>- Rear cleaning nozzle;<br>- Adjustable jet position;<br>- Emergency flushing system during power outages. |

The water–energy nexus in water-using products (WuPs) was also considered in the recent Unified European Water Label (UWL). According to studies by the European Commission (EC), replacing standard domestic devices (taps, showers, etc.) with water-efficient products will result in an overall reduction in annual domestic water consumption of up to 35% for taps and 11% for showers and 30% in associated energy. It should be noted that the Aveiro study is in line with these estimates [71].

The importance of these numbers is evident, considering that the total water withdrawal for use in taps and showers was estimated in the European Union (EU) to be around 25,000 Mm$^3$ in 2010, and the total energy demand associated with the use of taps and showers in EU countries was estimated at around 3000 PJ$^3$/year [72]. The total $CO_{2eq}$ emissions related to annual EU primary energy demand from taps and showers were estimated to be around 160 Mton in 2010.

However, strong EU policies for water efficiency in buildings have not been promoted in the past, which is why several efficiency labels for water-using products (WuPs) have emerged in different European countries in recent years, generally for voluntary use, as is the case with the ANQIP label shown in Figure 3 [73,74]. In 2018, an agreement was reached with representatives from ANQIP, Water Label, Swedish Label, and Swiss Label to develop a unified European label for all WuPs. This agreement resulted in the creation of the Unified Water Label (UWL), which is represented in Figure 4.

The Unified Water Label Association (UWLA), the entity responsible for developing the UWL, considered that a single label covering water and energy efficiencies would be the best option for the market. Therefore, in addition to a color grid associated with water consumption, the UWL also has an energy icon on the base, which can be seen in Figure 4, to make consumers aware of the predicted annual energy consumption when using the product.

Energy consumption associated with water flow rates or volumes (water–energy nexus) is not yet considered in European Standards, so the UWLA has established a calculation based on basic physics principles:

$$E = m \times C \times \Delta T \tag{1}$$

where
  *E* = energy [kWh];
  *m* = mass [kg];

$C$ = specific heat of water = 0.00116 kWh/(kg K);
$\Delta T$ = temperature difference [K].

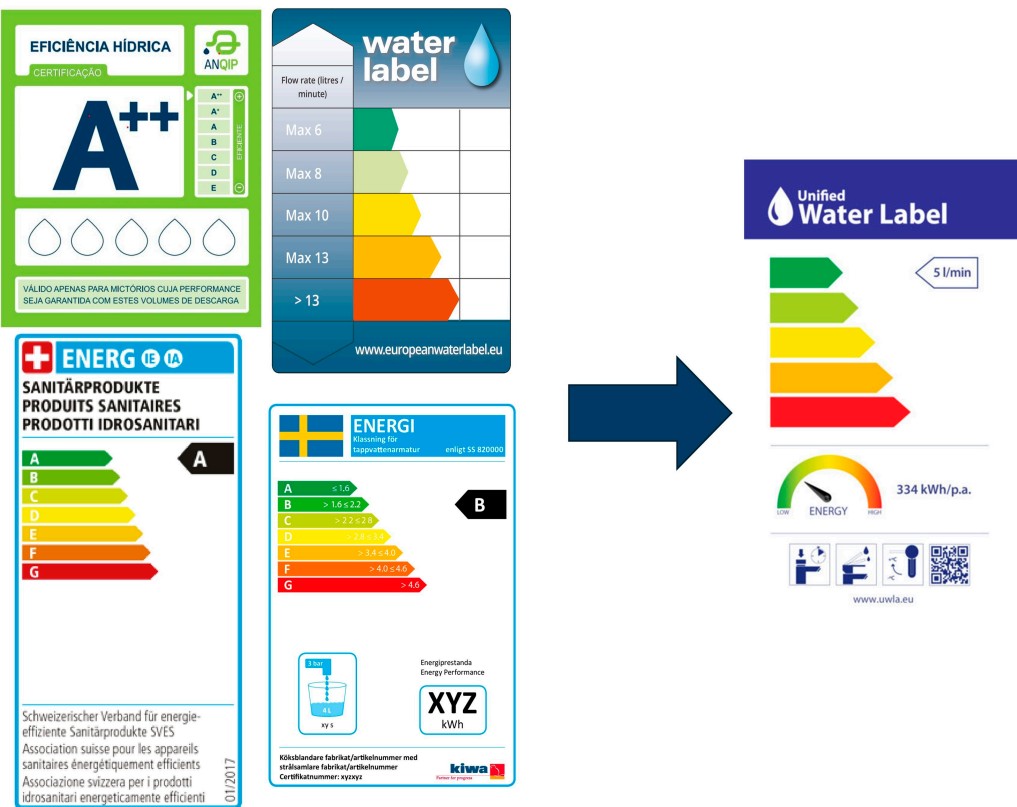

**Figure 4.** European Unified Water Label (UWL) [36].

With this basic calculation, together with average usage times, the expected annual energy consumption can be easily estimated. The average usage times to consider are as follows:

(1)    For washbasin taps (and bidets): 1 min per use and five uses per person and per day;
(2)    For kitchen taps: 1 min per use and five uses per person and per day;
(3)    For showers: 7 min per use and one use per person and per day.

With regard to washbasin taps and showers, an average outlet temperature of 38 °C is accepted, while for kitchen taps, the average outlet temperature is considered to be 45 °C. In all cases, the average inlet temperature is set at 15 °C. The main calculation assumptions were taken directly from an EC study on taps and showers [73,74]. For bathtubs, the same calculation can be used to raise awareness of the energy costs associated with filling the bathtub.

The development of rainwater-harvesting systems and the reuse of gray water in buildings are constructive solutions that have had significant dissemination in several countries over the last few years. Therefore, the analysis of the water–energy nexus for these installations is also of interest.

Rainwater-harvesting systems in buildings obviously reduce water (and energy) consumption from public networks, as rainwater can be used for purposes that do not require drinking water. However, they often require a pressurization system in the building.

It can easily be seen that the energy consumption when these systems are installed is equal to or lower than that which occurs when the supply is made via the public network, given that the pressurization systems are dimensioned for the minimum pressures required in the building, which is not the case with the public network, where pressures seek to respond to the general needs of the entire urban area.

With regard to the use of graywater, it can be said that so-called compact installations, with a short retention period and simplified treatment (as is the case with devices available on the market that combine a washbasin and flush toilet), also lead to savings in energy. In fact, there is a reduction in water consumption from the public network in these situations, with an inherent reduction in energy consumption, as previously mentioned.

In the case of installations with long retention periods, with "conventional" treatment, the energy consumed in the treatment process makes the system "neutral" from an energy point of view. In fact, the energy required to treat graywater, around 1.8 kWh/m$^3$, is approximately equivalent to the energy saved in the urban water cycle. (In the study carried out by ANQIP in the city of Aveiro, energy consumption in public networks was precisely determined to be around 1.8 kWh/m$^3$ [71]).

## 4. Conclusions

The water–energy–food nexus, to which some authors add climate, is the fundamental link for the sustainable development of humanity and human security. This nexus is valid globally but can also be considered on a regional or local scale or even applied in non-agricultural environments, such as cities or even buildings. In the latter cases, this nexus reflects the relationship between energy, water, and nutrients (or fertilizers).

With regard to the connection between water and nutrients, it should be noted that the recovery of phosphorus in buildings or urban environments will probably be a solution of the greatest relevance in the future. Phosphorus is a non-renewable element essential for food production, the lack of which can compromise humanity's food security, with its loss through urine being one of the main causes of losses in the value chain. Without prejudice to other applicable recovery technologies, for example in WWTPs, the installation of urine-diverting toilets and urinals in residential buildings and the generalization of urinals for women, for example, will probably be advantageous solutions for recovery in terms of energy and efficiency. The relationship between energy and nutrients in buildings is generally not very relevant. However, the global water–energy–nutrients nexus can be found in buildings at the level of some modern toilet composting solutions.

The relationship between water and energy is always very significant, and buildings are no exception. In addition to the important and well-known dependence that exists on the production of domestic hot water, there are many other aspects in which this connection is revealed in urban environments and buildings. This is the case, for example, for electricity consumption in public supply and drainage networks, which must, naturally, be associated with water consumption in buildings, and energy consumption resulting from pressurization needs in buildings or the existence of vacuum drainage systems. Furthermore, the rapid evolution in recent decades in the design of sanitary installations, seeking to increase comfort, hygiene, or sustainability parameters, has implied, in many situations, an associated energy consumption. Water labelling for water-using products (WuPs) must be encouraged, as, in addition to being an incentive for the installation of efficient devices, it in many cases incorporates information on the water–energy nexus, as is the case in the new European Unified Water Label (UWL).

Overall, the water–energy–nutrients nexus in urban environments and buildings is particularly relevant and, in some respects, may be essential for sustainable development and human food security. It should, therefore, be the subject of special attention, in parallel with the global and essential approach to the water–energy–food nexus.

**Author Contributions:** Conceptualization, A.S.-A. and C.P.-R.; methodology, A.S.-A.; validation, C.P.-R.; formal analysis, A.S.-A.; investigation, A.S.-A. and C.P.-R.; writing—original draft preparation, A.S.-A.; writing—review and editing, C.P.-R.; visualization, A.S.-A. and C.P.-R.; supervision, A.S.-A. All authors have read and agreed to the published version of the manuscript.

**Funding:** This research received no external funding.

**Data Availability Statement:** Data are contained within the article.

**Conflicts of Interest:** The authors declare no conflicts of interest.

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
