# Peer review of "Water–Energy–Nutrients Nexus of Urban Environments"

_water, doi:10.3390/w16060904_

Round 1

Reviewer 1 Report (New Reviewer)

Comments and Suggestions for Authors

Review of water-2886937

This is a unique review about the water-energy-nutrients nexus (and to some extent, water-energy-food nexus, waste-food-energy nexus) related to buildings, toilets, etc, which are not fully understood yet. Because of this uniqueness, then this manuscript shows some fresh multidisciplinary perspectives of environmental science, urban development science, chemistry, physics, energy, etc.

1.       Line 150-154: The extraction of phosphorus from industrial effluent, especially for domestic effluent (the focus of this manuscript, about nutrient from the toilets) is very interesting. Please make a comprehensive table about the extraction process of P, such as name of the process, the name of the equipment, P concentration in the effluent, P concentration in the final product, energy consumption, flow rate, time needed, etc.. For the algae-assisted P extraction process, please also write the name of algae. This table is important to show the added value of your manuscript for the scientific community. At this moment, the manuscript is dominated by qualitative descriptions. A comprehensive table with quantitative information is therefore urgently needed.

2.       Line 128-131: This paragraph tells about the importance of N, P, and K. However, the authors highlight the importance of discussing P in the next paragraph 132-139. As those macronutrients are equally important, then at the very least, please refer to these references that show the importance of NITROGEN from wastes (food, animal waste (manure), wastewater, etc.) in the perspective of water-food-energy nexus and/or waste-food-energy nexus:

  • Sustainability 2022, 14(21), 13993; https://doi.org/10.3390/su142113993 --> larvae of an edible insect to convert food waste and/or manure to be protein (indirectly produced as animal feed) and biodiesel (from the fat of the larvae) (waste-food-energy nexus)
  • Journal of Cleaner Production 2021, 299, 126861 https://doi.org/10.1016/j.jclepro.2021.126861 --> life cycle analysis of recycling of cassava wastewater stream for the study of water-waste-energy-food (WWEF) nexus via anaerobic digester process.
  • AQUA – Water Infrastructure, Ecosystems, and Society 2021, 70(2), 138–154 https://doi.org/10.2166/aqua.2020.058 --> summarizes the environmental issues associated with the water-energy-food nexus, especially from fertilizer, food waste and manure.

3.         Please also refer these references that show the importance of POTASSIUM (in the lithosphere and water stream) in the perspective of water-food-energy nexus and/or waste-food-energy nexus:

  • Sustainability 2022, 14(3), 1799; https://doi.org/10.3390/su14031799 --> water-food-energy nexus related to NPK fertilizer (20% N, 30% P, and 50% K)
  • Energy Nexus 2023, 2023, 10, 100207 https://doi.org/10.1016/j.nexus.2023.100207 --> Water-food-energy-ecosystem nexus about N, P, K. It also has the data of mass balance (of N, P, K, Ni, Zn, Cd) in lithosphere.

4.       Figure 1: Please provide the figure with a higher resolution. Please also show that you have the permission (from the publisher) to use the figure.

5.       Figure 2: Please also show that you have the permission (from the publisher) to use the figure.

6.       Line 169: Is it possible to revise to “…from approximately pH= 6… to pH= 9 after…”?

7.       Line 227: Please provide references to the amount of the energy consumption (8 MJ, 34 MJ).

8.       Line 298: Please provide the reference number to the “Aveiro study”.

9.       Line 303: Please provide the definition of “PJ3”. Is it cubic pico-Joule? But this unit is quite strange. Please check.

10.   Line 307: The abbreviation “WuP” has appeared for the first time in line 282. Please revise.

11.   Line 311: Please change the double fullstop to be single fullstop.

12.   Line 366: m3 --> superscripted 3.

13.   Reference 38: 2nd ed. --> add superscripted nd

14.   Reference 41: please write scientific name(s) in italics --> Scenedesmus obliquus

15.   Reference 63: 6th --> superscripted th.

Author Response

Reviewer 2 Report (New Reviewer)

Comments and Suggestions for Authors

The presented paper brings an interesting subject; however, it needs some improvements. Beginning with the title, i tis not adjusted to the content! It should have something about labelling that occupies part of the paper.

The novelty that the paper bring is not clear!

As a review paper it is not properly organized. For example, in Materials and Methods section it should be described how the literature review was done, with which keywords…

Figure 1: Image quality must be improved.

Line 149: there are technologies that allow phosphorous removal from wastewater treatment plants.

Lines 157-162: there is a contradiction in this paragraph. On one hand it is stated that urine recovered from buildings may be directly used as a fertilizer but on the other, in the second part of the sentence, it says it can’t.

Line 172: it really depends on the type and age of green roofs.

Line 178: references missing.

Line 216: please describe Aquatron system…

Lines 225-231: how much energy is spent in nutrients recovery in buildings? Without these, comparisons should not be made!

Line 233: Which studies? References missing

Line 311: Remove the second point (WuP..)

Conclusions are very general, they are not supported by the work developed in the paper.

Comments on the Quality of English Language

Needs improvement.

Round 2

Reviewer 1 Report (New Reviewer)

Comments and Suggestions for Authors

Review of water-2886938-v2 Thank you for the effort on revising the manuscript. It can be accepted now.

Note: There are tiny revisions that can be performed during the proofreading stage, as follows:
-Line 230: Please revise "[64,68]" to be "[64-68]".
-Title: Gramatically, perhaps the title should be written as "Nexus of..."

Reviewer 2 Report (New Reviewer)

Comments and Suggestions for Authors

Nothing to declare.

This manuscript is a resubmission of an earlier submission. The following is a list of the peer review reports and author responses from that submission.

Round 1

Reviewer 1 Report

Comments and Suggestions for Authors

The topic proposed in this manuscript is interesting although to suggest  quriosity and interest to readers it needs to include more specific references to previous works or related projects that might have been or that are qurrently  being developed. Another relevant fault found is that more than 70% of the references used are prior to 2018. An  important update should be considered. The introduction and results should include information on the extent of nutrient recovery in urban wastewater, what technologies are being actually used in treatment plants.

Comments on the Quality of English Language

Extensive editing of English is required

Reviewer 2 Report

Comments and Suggestions for Authors

The present paper present a integration of the results obtained in different studies, intending to highlight the strong connections between water, energy, and nutrients at the level of buildings.

I propose this study for publication taking into account  the following:

- the introduction provide sufficient background and iclude all relevant references;

- the results are clearly presented;

-the conclusion are suported by the results;

Reviewer 3 Report

Comments and Suggestions for Authors

Enclosed a file with a list of minor changes required for misprints,, unclear points etc..

Comments on the Quality of English Language

The quality of English is good as a whole.